# E3 Ligase ITCH Interacts with the Z Matrix Protein of Lassa and Mopeia Viruses and Is Required for the Release of Infectious Particles

**DOI:** 10.3390/v12010049

**Published:** 2019-12-31

**Authors:** Nicolas Baillet, Sophie Krieger, Xavier Carnec, Mathieu Mateo, Alexandra Journeaux, Othmann Merabet, Valérie Caro, Frédéric Tangy, Pierre-Olivier Vidalain, Sylvain Baize

**Affiliations:** 1Unité de Biologie des Infections Virales Emergentes, Institut Pasteur, 69365 Lyon, France; nicolas.baillet@hotmail.fr (N.B.); sophieeve@live.fr (S.K.); xavier.carnec@pasteur.fr (X.C.); Mathieu.mateo@pasteur.fr (M.M.); Alexandra.journeaux@pasteur.fr (A.J.); Othmann.merabet@pasteur.fr (O.M.); 2Centre International de Recherche en Infectiologie, Université Claude Bernard Lyon I, INSERM, CNRS, ENS Lyon, 69007 Lyon, France; 3Unité Environnement et Risques Infectieux, Cellule d’Intervention Biologique d’Urgence, Infection et Epidémiologie, Institut Pasteur, 75015 Paris, France; valerie.caro@pasteur.fr; 4Unité de Génomique Virale et Vaccination, Institut Pasteur, CNRS UMR-3569, 75015 Paris, France; Frederic.tangy@pasteur.fr (F.T.); pierre-olivier.vidalain@inserm.fr (P.-O.V.); 5Biologie Cellulaire des Infections Virales, Centre International de Recherche en Infectiologie, Université Claude Bernard Lyon I, INSERM, CNRS, ENS Lyon, 69007 Lyon, France

**Keywords:** Lassa fever, LASV, MOPV, ITCH, Z matrix protein, yeast two-hybrid screening, egress, infection

## Abstract

Lassa virus (LASV) and Mopeia virus (MOPV) are two closely related, rodent-born mammarenaviruses. LASV is the causative agent of Lassa fever, a deadly hemorrhagic fever endemic in West Africa, whereas MOPV is non-pathogenic in humans. The Z matrix protein of arenaviruses is essential to virus assembly and budding by recruiting host factors, a mechanism that remains partially defined. To better characterize the interactions involved, a yeast two-hybrid screen was conducted using the Z proteins from LASV and MOPV as a bait. The cellular proteins ITCH and WWP1, two members of the Nedd4 family of HECT E3 ubiquitin ligases, were found to bind the Z proteins of LASV, MOPV and other arenaviruses. The PPxY late-domain motif of the Z proteins is required for the interaction with ITCH, although the E3 ubiquitin-ligase activity of ITCH is not involved in Z ubiquitination. The silencing of ITCH was shown to affect the replication of the old-world mammarenaviruses LASV, MOPV, Lymphocytic choriomeningitis virus (LCMV) and to a lesser extent Lujo virus (LUJV). More precisely, ITCH was involved in the egress of virus-like particles and the release of infectious progeny viruses. Thus, ITCH constitutes a novel interactor of LASV and MOPV Z proteins that is involved in virus assembly and release.

## 1. Introduction

Viral hemorrhagic fevers (VHFs) caused by mammarenaviruses represent a serious and increasing public health concern [1]. Arenavirus-associated VHFs are zoonotic diseases, with humans acting as accidental hosts after contact with a viral reservoir, typically rodents [2]. The old-world (OW) arenavirus complex known to cause VHF includes Lassa virus (LASV) and the recently discovered Lujo virus (LUJV) [3,4]. The prototypic arenavirus, lymphocytic choriomeningitis virus (LCMV), which is distributed worldwide, is also of clinical significance [5]. The most prevalent pathogen among the arenaviruses is LASV, the causative agent of Lassa fever (LF), which is a significant cause of morbidity and mortality, with tens of thousands of cases annually and thousands of fatalities in West Africa [6]. The disease is characterized by acute forms associated with fever, myalgia, abdominal pain, nausea, diarrhea, cough, sore throat, and facial edema and evolves toward a shock syndrome in the terminal stage for severe cases. The tendency of LASV to cause outbreaks has steadily risen in Nigeria over the last three years, with laboratory-confirmed cases increasing from 106 in 2016 to 143 in 2017 and reaching 562 by November 2018 [7]. Additionally, there is currently no licensed vaccine or efficient antiviral drug available in the field against this disease, for which the area of endemicity is expanding [8]. The particularity of arenaviruses is the presence of non-pathogenic agents that are not associated with human disease, despite their isolation from the same host species. This is true for Mopeia virus (MOPV), which is closely related to LASV but has never been associated to human infection [9]. MOPV has even been shown to protect non-human primates against a challenge with LASV, showing that both viruses are antigenically related. Comparing LASV and MOPV should therefore allow the identification of immune and viral features involved in LF pathogenesis and, to a lesser extent, other arenavirus-associated VHFs.

The bi-segmented RNA genome of the Arenaviridae family encodes four proteins: the nucleoprotein (NP), the surface glycoprotein precursor (GPC), the L polymerase, and the RING finger protein Z [10,11]. Although it is the smallest arenavirus protein, the Z matrix protein plays multiple functions in the viral life cycle, including structural and non-structural roles. The Z protein is known to regulate key host-cell functions through direct interactions to favor viral replication. These cellular factors include the promyelocytic leukemia protein (PML), the nuclear fraction of the ribosomal protein P0, the eukaryotic translation initiation factor 4E (eIF4E), and the proline-rich homeodomain protein (PRH) [12,13,14]. The Z protein may also be involved in viral escape from the innate immune response [15,16]. Previous studies have also revealed an inhibitory effect of the Z protein on both transcription and replication using a LCMV reverse genetic system [17]. In addition to its non-structural roles, the Z protein plays a critical role in viral assembly by mediating the incorporation of GP, NP, and the L polymerase into nascent virions [17,18,19,20,21,22,23,24]. Importantly, the Z proteins of LASV and LCMV contain a PPxY late domain that is known, as for enveloped viruses, to mediate viral budding through the recruitment of the ESCRT-associated protein TSG101 and Alix [25,26]. The LASV Z PPxY motif has also been shown to interact with Nedd4, from the Nedd4 family of HECT E3 ligases and this interaction may be required for efficient budding of the virus, as has been shown during various viral infections [27,28,29,30,31,32,33]. Overall, the arenavirus Z protein is central in the viral life cycle and interacts with a variety of cellular factors that may have been only partially discovered and are not yet fully understood.

We addressed this potential knowledge gap by screening the Z proteins from non-pathogenic MOPV and pathogenic LASV Z protein interactions using high-throughput yeast two-hybrid screening (Y2H). Twenty-four host-cell partners were identified and validated in a secondary Y2H assay, including two Nedd4-family proteins, ITCH and WWP1. We therefore focused on the functional role of these proteins during non-pathogenic MOPV and pathogenic LCMV, LASV, and LUJV infections. We observed that ITCH was required for the replication of tested viruses, and confirmed with other techniques the interaction of ITCH with the Z proteins of different arenaviruses. We showed that the interaction of MOPV and LASV Z with ITCH occurred through their PPXY late domain and did not require the E3 ubiquitin-ligase activity of ITCH. Finally, we demonstrated that ITCH was important for LASV and MOPV infectious-particle production and was required for the release of viral progeny.

## 2. Materials and Methods

### 2.1. Yeast Two-Hybrid Screening

Y2H screens were performed following the protocol described in Vidalain et al. [34]. DNA sequences encoding the matrix proteins (Z) of LASV or MOPV were cloned by in vitro recombination (Gateway technology; Invitrogen, Carlsbad, CA, USA) from pDONR207 into the Y2H vector pPC97-GW (kindly provided by Dr. Marc Vidal) for expression in fusion downstream of the GAL4 DNA-binding domain (GAL4-BD). AH109 yeast cells (Clontech; Takara, Mountain View, CA, USA) were transformed with these baits using a standard lithium acetate protocol. Spontaneous transactivation of the *HIS3* reporter gene was observed in yeast cells expressing GAL4-BD-Z. Consequently, screens were performed on a synthetic medium lacking histidine (-His) and supplemented with 3-amino-1, 2, 4-triazole (3-AT) at 5 to 10 mM for MOPV and 80 to 100 mM for LASV. A mating strategy was used to screen three different prey libraries with distinct characteristics: a human spleen cDNA library (Invitrogen), a mouse brain cDNA library (Invitrogen), and a normalized library containing 12,000 human ORFs (CCSB Human ORFeome, Open Biosystems, Dharmacon, Lafayette, CO, USA). All libraries were established in the Y2H plasmid pPC86 to express prey proteins in fusion downstream of the GAL4 transactivation domain (GAL4-AD). After six days of culture, colonies were picked and replica plated over three weeks on selective medium to eliminate potential contamination with false positives. Prey proteins from selected yeast colonies were identified by PCR amplification using primers that hybridized within the pPC86 regions flanking the cDNA inserts. PCR products were sequenced and cellular interactors identified by multi-parallel BLAST analysis (kindly performed by Louis M. Jones, Institut Pasteur, Paris, France).

### 2.2. GAP-Repair Procedure

PCR products were co-transformed into AH109 yeast cells expressing GAL4-BD-Z constructs together with an empty pPC86 vector linearized downstream of the GAL4-AD coding sequence [35]. Homologous recombination in yeast cells between PCR products and linearized pPC86 vectors allows the reconstruction of GAL4-AD-Prey sequences. Transformed cells were plated on selective -His media supplemented with 3AT at 5 mM for the MOPV-Z protein, and 100 mM for the LASV-Z protein. After five days of culture on selective medium, growing colonies were scored.

### 2.3. Cell Lines and Viruses

VeroE6 cells were maintained in Dulbecco’s Modified Eagle Medium (DMEM) supplemented with 0.5% penicillin-streptomycin (PS) and 5% fetal bovine serum (FBS). The A549, HeLa, and HEK-293T cell lines were maintained in DMEM with 0.5% PS and 10% FBS. LCMV (WE strain), LUJV (GenBank accession number NC_012776), MOPV (AN21366 strain [9], GenBank accession numbers JN561684 and JN561685), and LASV (AV strain [36], GenBank accession numbers FR832711 and FR832710) viruses were grown in Vero E6 cells at 37 °C in 5% CO_2_. Viral supernatants were harvested and used as the virus stock and the absence of mycoplasma was confirmed. LASV, LCMV, LUJV, and MOPV titers were determined by plaque immunoassays as described below. All experiments with LASV, LCMV, and LUJV were carried out in biosafety level 4 facilities (Laboratoire P4 Jean Mérieux—INSERM, US003, Lyon, France). Recombinant MOPV-WT and MOPV with a FLAG-tagged Z (MOPV-ZF) protein was obtained by reverse genetics as described here [37].

### 2.4. Plasmids, Antibodies, and Reagents

The Z ORFs of MOPV (ZMop), LASV (ZLas), LCMV (ZLcm), LUJV (ZLuj) and JUNV (ZJun) are cloned in the pHCMV-MCS between the HindIII and BamHI sites and carried a C-terminal FLAG. ZM-mCherry and ZL-mCherry fusion protein were cloned in the pHCMV-MCS between the XmaI and BamHI sites with the mCherry fluorescent tag in the C-terminal position. Alanine mutants of the LASV and MOPV Z protein were obtained by alanine-scanning mutagenesis from the ZM-FLAG or ZL-FLAG vectors, in which five amino acids were mutated into alanine repeatedly along the entire sequences. Directed mutagenesis was performed with the QuickChange Site-Directed Mutagenesis Kit II (200524, Agilent, Santa Clara, CA, USA), according to the manufacturer’s instructions. HA-Ubiquitin (HA-Ub) and HA-Ubiquitin KO (HA-Ub-KO) (in which all lysine residues are mutated into arginine) plasmids were kindly provided by C. Journo (CIRI, Lyon, France) and have been previously described [38]. The eGFP-ITCH construct was kindly provided by Y. Jacob (Unité de Génétique Moléculaire des Virus à ARN, Institut Pasteur, Paris, France). ITCH-C830A was obtained by site-directed mutagenesis from the ITCH-WT construct. eGFP-ITCH-WT and eGFP-ITCH-C830A were engineered into peGFP-C1, allowing the expression of GFP-tagged proteins for immunofluorescence experiments. For the immunoprecipitation assays using the ITCH-HA plasmid, ITCH with a C-terminal HA tag was cloned in the phCMV-MCS plasmid. All plasmid constructs were verified by sequencing.

The primary antibodies used were anti-β actin (1:20,000; A3854) and anti-FLAG (1:3000; A8592), both from Sigma-Aldrich (St Louis, MO, USA); anti-HA (1:2000; ab9110), anti-WWP1 (1:500; ab43791) and anti-WWP2 (1:1000; ab103527) from Abcam (Cambridge, MA, USA); anti-GAPDH (1:11,000; sc-25778) from Santa Cruz Biotechnology (Dallas, TX, USA), anti-Ubiquitin (1:1000) from R&D systems (Noyal Chatillon Sur Seiche, France) and anti-ITCH (1:500; 12117) from Cell Signaling Technology (Beverly, MA, USA). The secondary antibodies used for western blotting were anti-mouse conjugated to peroxidase (1:20,000; 111-035-174) and anti-rabbit conjugated to peroxidase (1:20,000; 111-035-144), both from Jackson ImmunoResearch (Cambridge House, UK).

### 2.5. RNAi Analysis

For all the experiments, 293T, HeLa, or A549 cell lines were reverse transfected with the indicated siRNA at a final concentration of 20 nM using Lipofectamine RNAiMAX (Invitrogen), according to the manufacturer’s instructions. The cells were maintained for 72 h after siRNA transfection. For the analyses of the effect of ITCH depletion on viral replication, HeLa or A549 cells were counted and infected with MOPV or LASV, at an MOI of 0.1 for HeLa or 0.01 for A549, for 1 h in DMEM medium supplemented with 2% FBS and 0.5% PS. Infectious medium was removed before adding fresh medium (DMEM 5% FBS, 0.5% PS) at indicated time points. Viral RNA and infectious particles were then quantified (please see corresponding paragraph). All siRNA targeting ITCH, WWP1 or WWP2 as well as the non-targeting control siRNA (smart pools si-GENOME) came from Dharmacon.

### 2.6. Coimmunoprecipitation and Western-Blot Analysis

Human 293T cells were transfected for 15 h with the indicated plasmids using Lipofectamine 2000 (Invitrogen), according to the manufacturer’s protocol. For coimmunoprecipitation assays during infection, A549 cells were infected with MOPV-ZF for 48 h. The 293T and A549 cells were harvested and lysed in non-denaturing lysis buffer composed of 10 mM Hepes, 5 mM EDTA, 150 mM NaCl, 1% NP40, and 50 µM PR-619 (662141, Calbiochem, Merck, Darmstadt, Germany) and supplemented with protease inhibitors (Sigma-Aldrich, 11873580001). Lysates were clarified for 15 min at 10,000 rpm. Supernatants were then incubated with M2 anti-FLAG magnetic beads (M8823, Sigma) for 2 h at 4 °C under agitation, or with anti-HA magnetic beads (88836, Thermo Fischer Scientific, Waltham, MA, USA) for 2 h at room temperature. The beads were then washed four times in lysis buffer before boiling 15 min in loading buffer. For western blots, equal amounts of protein were loaded and separated on 4 to 15% gradient precast gels and transferred onto Polyvinylidene fluoride (PVDF) membranes before staining. Samples were immunoblotted with the appropriate primary antibodies. Protein levels were quantified by measuring the intensity of the bands by densitometry. Actin or GAPDH were used as positive controls of the cell extracts.

### 2.7. Microscopic Analysis

All images were acquired using a confocal Zeiss LSM 510 (Zeiss, Oberkochen, Germany) with an Axioscope 63× oil immersion lens objective. The images were analyzed using ImageJ/Fiji (version 1.52 m). HeLa cells were cultured in 12-well plates with a sterile coverslip in each well. The cells were fixed in 4% paraformaldehyde for 20 min and treated with glycine (0.1M). All samples were mounted with DAPI (P36931, Invitrogen).

### 2.8. VLP Budding Assays

The 293T cells were transfected with si-ITCH or si-NT as described above (RNAi analysis). ITCH-silenced or control 293T cells were then transfected with 2 µg ZL-FLAG, ZM-FLAG, or empty phCMV vector using Lipofectamine 2000. At 15 h post transfection, the supernatants were harvested and the cells lysed for analysis of whole cell extracts and immunoprecipitation by western blotting. Supernatants were clarified for 15 min at 5000 rpm. Cleared supernatants were then deposited onto a sucrose cushion (PBS—20% sucrose) before ultracentrifugation at 56,000 rpm for 90 min (TW60 rotor, Optima L-100 XP ultracentrifuge, Beckman Coulter, Villepinte, France). Pellets were resuspended in 50 µL loading buffer before analysis by western blotting.

### 2.9. Quantitative Viral RNA Analysis

Viral RNA was extracted from culture supernatants and cells using the QIAamp Viral RNA and RNeasy Minikits, respectively (all from Qiagen, Hilden, Germany), according to the manufacturer’s instructions. Quantitative PCR for viral RNA was performed with the EuroBioGreen Lo-ROX qPCR mix (Eurobio, Les Ulis, France), using primers 5′-CTTTCCCCTGGCGTGTCA-3′ and 5′-GAATTTTGAAGGCTGCCTTGA-3′ for MOPV and 5′-CTCTCACCCGGAGTATCT-3′ and 5′-CCTCAATCAATGGATGGC-3′ for LASV.

### 2.10. Virus Titration

Vero E6 cells were infected with sequential dilutions of supernatant and incubated at 37 °C in 5% CO_2_ for six days with Carboxy-methyl-cellulose (1.6%) (BDH Laboratory Supplies, Poole, UK) in DMEM supplemented with 2% FBS. Infectious foci were detected by incubation with monoclonal antibodies (mAbs) directed against MOPV, LASV, LCMV, or LUJV (mAbs L52-54-6A, L53-237-5, and YQB06-AE05, generously provided by P. Jahrling, USAMRIID, Fort Detrick, MD), followed by PA-conjugated goat polyclonal anti-mouse IgG (Sigma-Aldrich).

### 2.11. Flow Cytometry

A549 cells were transfected with non-targeting siRNA or ITCH targeting siRNA at a final concentration of 30 nM for 72 h before infection with recombinant MOPV-WT or MOPV-ZFLAG at a MOI of 2 for 24 h. Cells were then harvested, washed, and the pellets suspended 20 min in PBS with 4% formaldehyde. Cells were then permeabilized with PBS complemented with 5% FBS, 0.5% saponin (47036-50G-G, Sigma), 0.02% sodium azide (S-8032, Sigma) before being incubated with anti-FLAG-APC antibody (130-101-565, Miltenyi Biotec, Paris, France) for 30 min at 4 °C. The fluorescence of formaldehyde-fixed cells was measured using a BD LSR II flow cytometer (BD Biosciences, San Jose, CA, USA). Data were analyzed using Kaluza software (Beckman Coulter, version 1.2).

### 2.12. Statistical Analysis

Statistical analyses were performed using GraphPad Prism (version 8.0.2, GraphPad Software, San Diego, CA, USA). The distribution of samples was analyzed by the Shapiro–Wilk test before using non-parametric (Mann–Whitney or Kruskal–Wallis) tests to determine the significance of differences between groups (* *p* < 0.05, ** *p* < 0.01, *** *p* < 0.001, n.s., non-significant).

## 3. Results

### 3.1. Identification of Host-Cell Interactors for the LASV and MOPV Z Matrix Proteins

We identified host-cell proteins that interact with the Z protein of both viruses using a total of six yeast two-hybrid (Y2H) screens. Yeast cells expressing the Z protein of LASV or MOPV fused to GAL4-DB domain (viral baits) were used for screening three different libraries expressing either human or mouse proteins fused to GAL4-AD domain (cellular preys). Interactions between bait and prey proteins lead to transactivation of the *HIS3* reporter gene. After selection on culture medium lacking histidine, prey proteins from positive yeast colonies were identified by PCR amplification, sequencing and multi-parallel BLAST analysis. We then confirmed the positive candidates obtained in the Y2H system by retesting each interaction pairwise in fresh yeast cells (gap repair procedure, see Material and Methods). A total of twenty-four host cell partners were identified and are listed in Table 1. The interactions between TSG101 or PDCD6IP (ALIX) and arenavirus Z protein have already been reported by others and validate our screen [26,33]. The functional classification of the various Z-protein interactors revealed the presence of six genes related to autophagy pathways, the role of which we have already reported during LASV and MOPV infection [39]. The screens revealed several other host-cell partners, mainly involved in phosphorylation or the addressing of proteins, as well as membrane trafficking. The role of these proteins during arenavirus infection is currently under investigation by our group. Interestingly, the screens revealed that ITCH and WWP1 interact with both the LASV and MOPV Z proteins. These proteins belong to the Nedd4 family of E3 Ubiquitin ligases and may play an important role in the life cycle of MOPV and LASV (family reviewed in reference [40]).

### 3.2. ITCH Is Required for Efficient Old-World Arenavirus Infection

We determined the functional relevance of ITCH and WWP1 to old-world arenavirus infection by investigating the effect of siRNA-mediated depletion of these proteins during infection by four old-world arenaviruses, including three that are pathogenic (LCMV, LASV, LUJV) and one that is not (MOPV). As WWP1 and WWP2 share a high degree of homology among the Nedd4 family of HECT E3 Ubiquitin ligases, we also tested the impact of WWP2 silencing on viral replication. A total of 72 h after transfection of the siRNA, human adenocarcinoma alveolar basal epithelial (A549) cells were efficiently depleted for ITCH, WWP1 or WWP2, as compared to cells transfected with a non-targeting (NT) control siRNA (Figure 1A). Silenced cells were then infected with each indicated virus at a multiplicity of infection (MOI) of 0.01 and incubated for 48 h. Titration of the supernatants showed ITCH to be required for LCMV, MOPV, and LASV infectious particle production, with a significant drop in viral release of >70% (Figure 1B). Although not statistically significant, the depletion of WWP1 and WWP2 moderately decreased infectious particles production for all assessed infections. LUJV infectivity in ITCH-depleted cells was greater than that in cells depleted for WWP1 or WWP2, with an approximate 70% inhibition in the number of infectious particles as compared to the control condition.

### 3.3. The LASV and MOPV Z PPxY Late Motif Binds to ITCH

As ITCH appeared to be a potent proviral factor among old-world arenaviruses, we next sought to confirm its interaction with various arenavirus Z proteins in mammalian cells. The 293T cells were transfected with a C-terminal tagged FLAG Z protein of MOPV (ZMop-FLAG), LASV (ZLas-FLAG), LCMV (ZLcm-FLAG), LUJV (ZLuj-FLAG), or the additionally added new-world arenavirus JUNV (strain Candid#1) (ZJun-FLAG). Cells were co-transfected with C-terminal HA fused ITCH (ITCH-HA) and after 15 h of culture, cell lysates were either directly harvested or immunoprecipitated with anti-HA- (Figure 2A) or anti-FLAG- (Figure 2B) coupled magnetic beads, followed by western-blot analysis of the coimmunoprecipitated partners of the FLAG- or HA-tagged proteins. We identified both the ZMop-FLAG and ZLas-FLAG proteins in the ITCH-HA precipitates (Figure 2A). Conversely, ITCH-HA coimmunoprecipitated with the ZMop-FLAG and ZLas-FLAG proteins (Figure 2B). Additionally, ZLcm-FLAG, ZJun-FLAG, and, to a lesser extent, ZLuj-FLAG also coimmunoprecipitated together during either the HA- or FLAG-mediated immunoprecipitation. These results confirm the interactions detected by Y2H and demonstrate that the interaction of the Z matrix protein with ITCH represents a common feature for both old-world and at least one new-world arenavirus.

We next investigated the functional role of the interaction with ITCH by focusing on two old-world arenaviruses, the pathogenic LASV and its nonpathogenic homolog MOPV. We assessed colocalization between the Z protein and ITCH in another mammalian cell line to acquire images of the previously identified interaction with ITCH. Briefly, HeLa cells were co-transfected with plasmids expressing a fluorescent Z protein of MOPV (ZMop-mCherry) or LASV (ZLas-mCherry) and those expressing fluorescent ITCH (eGFP-ITCH). After 15 h of culture, cells were fixed and analyzed by confocal microscopy. The images show the relocalization of ITCH, which transits from a diffused form inside the cells under the control condition to localized puncta near the Z protein aggregates of MOPV and LASV (Figure 2C). The Z proteins of MOPV and LASV colocalized with ITCH at these sites, consistent with the immunoprecipitation assays.

The Arenavirus Z matrix protein is known to interact with host-cell partners through proline-rich late domains (PT/SAP and PPxY motifs) [25,41]. Moreover, the PPxY motif of other viral matrix proteins has been shown to interact with E3 ubiquitin-ligase family members [31,42]. In addition, a recent study demonstrated that the PPxY domain of Ebola virus VP40 interacts specifically with ITCH [27]. Thus, it seemed likely that the MOPV and LASV Z late domains containing the PPxY motif could directly bind ITCH. We tested this hypothesis by alanine-scanning directed mutagenesis, generating mutants in which five amino acids were sequentially mutated into alanine along the entire length of the MOPV and LASV Z proteins (Figure 2D). The 293T cells were co-transfected with ITCH and the indicated FLAG-tagged Z mutants, most mutations being in the late domains (ZM-A17 to ZM-A21 and ZL-A16 to ZL-A20) (Figure 2E). Cell extracts were then incubated with anti-FLAG-coupled magnetic beads and the presence of ITCH was assessed in the immunoprecipitated samples by western blotting. ITCH was undetectable in the precipitates of cells transfected with ZM-A20 and ZM-A21 mutants of MOPV and the ZL-A19 and ZL-A20 mutants of LASV. These mutants carried alanine mutations inside the PPxY late domain (Figure 2D), thus showing that this late domain is essential for the interaction between ITCH and LASV or MOPV Z protein. Overall, our findings demonstrate that ITCH interacts with the PPxY late domain of the Z protein of various mammarenaviruses.

### 3.4. ITCH Is Not Involved in Z Ubiquitination

Because ITCH mainly plays a role in transferring ubiquitin to protein substrates, we assessed the impact of the loss of interaction between Z and ITCH on Z ubiquitination. First, 293T cells were transfected with non-targeting siRNA or si-ITCH and after two days, transfected with HA-Ub and the Flag-tagged MOPV (ZM) or LASV (ZL) proteins or the Flag-tagged mutant proteins unable to bind ITCH, ZM-A20 or ZL-A19. Twenty-four hours post transfection, cells were lysed or incubated with anti-FLAG magnetic beads before analysis of the ubiquitinated MOPV and LASV Z protein by western blotting (Figure 3A). We observed in cells transfected with HA-Ub and ZM or ZL that an important fraction of ubiquitinated proteins of approximately 34 kDa were co-immunoprecipitated with ZM-FLAG or ZL-FLAG, corresponding to mono-ubiquitinated forms of the LASV and MOPV Z protein. The HA signal was lower in cells transfected with mutants ZM-A20 and ZL-A19. We did not observe significant differences in the quantity of ubiquitinated forms of ZM or ZL between si-CTL and si-ITCH cells after immunoprecipitation. Thus, the late domain of MOPV and LASV Z protein is critical for Z ubiquitination and ITCH does not seem to be involved in this specific ubiquitination.

In a second experiment, we assessed whether the overexpression of ITCH-WT or an enzymatically inactive mutant of ITCH (ITCH-C830A) can affect the ubiquitination of the Z protein. 293T cells were transfected with a control plasmid or combinations of plasmids encoding HA-Ub, FLAG-tagged ZM or ZL, ITCH-WT, or the enzymatically inactive ITCH-C830A mutant. Cell lysates were immunoprecipitated with anti-FLAG magnetic beads and the presence of ITCH, Z, and HA-Ub were detected by western-blot analysis (Figure 3B). Both ITCH-WT and ITCH-C830A coimmunoprecipitated with MOPV and LASV Z protein. As compared to control, ITCH-WT overexpression increased the HA-Ub signal that coimmunoprecipitated with MOPV or LASV Z proteins whereas such an increase in HA-Ub signal was not observed with ITCH-C830A. We also noticed that a basal amount of ubiquitinated forms of Z was still present when the dominant inactive form ITCH-C830A was transfected. These data indicate that even if ITCH can promote the ubiquitination of MOPV or LASV matrix protein, it is likely not the sole E3 ubiquitin ligase responsible for such ubiquitination.

Finally, we assessed the impact of ITCH depletion on MOPV Z ubiquitination during infection. A549 cells were transfected with non-targeting siRNA or si-ITCH for 72 h and infected for two days with MOPV expressing a tagged-FLAG Z protein (MOPV-ZF, in duplicates). Cell extracts were then immunoprecipitated with anti-FLAG magnetic beads before analysis by western blotting (Figure 3C). While ITCH could interact with MOPV Z protein during infection, as shown by the slight quantity of ITCH co-immunoprecipitated with the Z protein in si-CTL cells, ubiquitinated Z protein could not be detected in the FLAG precipitates, in contrast with the previous observations in transfected cells.

### 3.5. ITCH Promotes LASV and MOPV Infectious Particle Production and Is Recquired for Viral Release

We further evaluated whether ITCH can affect the LASV and MOPV life cycles and which stage may be targeted using an RNAi approach. Briefly, A549 and HeLa cells were transfected with a non-targeting siRNA (si-NT) or a pool of ITCH-targeting siRNA (si-ITCH). The si-ITCH transfected A549 and HeLa cells showed markedly lower ITCH expression 72 h after transfection than those under control conditions (Figure 4A). The A549 and HeLa cells were then infected with LASV or MOPV during 48 and 72 h, respectively (corresponding to the peak of the infection in our experimental settings). We then measured the quantity of viral RNA and infectious particles in the supernatants and the quantity of viral RNA inside A549 and HeLa cell lines. There was no difference in the amount of intracellular viral RNA between the si-CTL and si-ITCH conditions for either LASV or MOPV in both cell types, showing that the viral entry and replication steps were not affected by the depletion of ITCH (Figure 4B). However, the quantity of viral RNA (Figure 4C) and infectious particles (Figure 4D) released into the supernatant of infected si-ITCH cells was significantly lower than that of the infected si-NT in both cell lines. Indeed, we observed a reduction in the production of infectious particles of approximately 90% for MOPV and LASV in si-ITCH infected A549 cells, and of approximately 80% for both viruses in si-ITCH infected HeLa cells, relative to control conditions.

We then further investigated which viral stages are particularly affected by ITCH by analyzing the egress of MOPV and LASV virus-like particles (VLPs). The expression of arenavirus Z protein in the absence of other viral proteins is sufficient for the release of Z VLPs that are morphologically similar to virus particles released from infected cells [43]. We evaluated the quantity of MOPV and LASV Z-mediated VLP release in the supernatant of ITCH-depleted or control cells transfected with a FLAG-tagged MOPV or LASV Z protein (Figure 4E). After ultracentrifugation of the supernatants, we observed significantly fewer MOPV and LASV VLPs for si-ITCH cells than those under control conditions. The reduction of VLP egress was similar for MOPV and LASV (approximately 50%), as shown in the right panel in which the levels of Z-VLPs have been normalized to the quantity of transfected Z protein inside the cells. Finally, we assessed by flow cytometry the amount of FLAG-tagged Z protein in si-CTL or si-ITCH cells infected with recombinant MOPV (MOPV-WT) or MOPV expressing a FLAG tagged Z protein (MOPV-ZF) (Figure 4F). At 24 h post infection, cells silenced for ITCH and infected with MOPV-ZF expressed higher amounts of intracellular Z protein than cells silenced with a non-targeting si-RNA and infected by the same virus. Overall, these results show that ITCH positively regulates the late stages of both LASV and MOPV infection, by promoting viral egress and increasing infectious particle production.

## 4. Discussion

Arenavirus Z protein is a critical regulator that mediates the budding steps prior to virus release, ensuring the packaging of all viral components required for infectious-particle production [17,18,19]. To this end, the Z protein recruits several host proteins to facilitate the efficient release of viral progeny from the plasma membrane of the infected cell. To date, lists of host-cell proteins associated with various arenavirus infections have been established, providing new bases to investigate the potential roles of such interactions during the course of infection. With the exception of one study, which focused on the proteins of cellular origin associated with the Z protein of JUNV, most reports have focused on the NP and L interactomes during JUNV and LCMV infection [44,45,46,47,48]. However, these reports were based on the expression of tagged viral proteins through transfection or infection with genetically modified viruses and the identification of associated host-cell partners after purification and mass spectrometry analysis. Large protein complexes can be analyzed as a whole with this approach but discriminating direct or proximal interactors from indirect or distant ones is usually difficult. Here, we used the Y2H assay to map cellular targets of the Z proteins from the pathogenic LASV and its non-pathogenic homolog MOPV. This approach is known for being complementary to protein complex analysis by mass spectrometry, and because it is performed in a heterologous expression system, the detection of direct protein-protein interactions is favored. This method, coupled with gap-repair validation, allowed us to isolate more than 20 proteins that interact directly with MOPV and/or LASV Z proteins, most of these interactions being novel, with no previous reports in the literature. From this list of interactors, we showed that ITCH favors infectious-particle production of various arenaviruses, including MOPV and LASV, by promoting the egress of viral progeny. Our data show that the interaction between LASV or MOPV Z protein and ITCH depends on the PPxY late domain. Overall, our results complete the existing list of arenavirus-host interactions and contribute to a better understanding of LF pathogenesis.

The Y2H assays could present a bias, as Z protein is known to localize with cell membranes. It is thus possible that some interactions between the Z protein and host-cell proteins that take place in specific subcellular membranous compartments are unlikely to be reproduced in the Y2H screen performed in this study, because the interactions had to occur in the yeast nuclei. However, it is expected that the Z protein interacts with multiple host factors, considering the major structural and non-structural roles played by this protein during infection. The various host-cell partners we identified by Y2H are involved in autophagy, suggesting that the Z protein could hijack this pathway during MOPV and/or LASV replication. More specifically, six autophagy-related proteins that interact with the MOPV Z protein previously led us to investigate the impact of this pathway on MOPV and LASV infection [39]. Similar to the results we describe here for ITCH, autophagy promotes the production of infectious particles during both MOPV and LASV infection in HeLa cells. In addition, we showed here that UBQLN family members (namely UBQLN 1, 2, and 4) interact with the Z protein of MOPV, highlighting the link between autophagy and arenavirus infection. Members of the UBQLN family are involved in the regulation of ubiquitinated proteins and their targeting towards autophagic degradation [49]. UBQLN4 has also been identified in a Y2H screening as a target of the nsP2 protein of Chikungunya virus (CHIKV), suggesting that this interaction could participate in the hijacking of autophagy to the benefit of viral replication, as shown for various RNA viruses, including CHIKV [50,51]. These findings underline the relevance of the hits identified in our screen. Consistent with this hypothesis, we also identified TSG101 and PDCD6IP (also known as ALIX), which are involved in intracellular vesicular trafficking and have already been identified as arenavirus Z partners by others [26,46]. Our Y2H screen also identified various proteins involved in protein processing, including the endogenous γ2 isoform of AP-1 (AP1G2), which has been shown to interact with both MOPV and LASV Z protein. Interestingly, a previous study demonstrated that the HIV-1 protein Nef interacts with AP1G2 in clathrin-coated pits to promote efficient internalization of its co-receptor CD4 and its subsequent targeting to lysosomes, therefore counteracting the detrimental effects of high CD4 expression in HIV-infected cells, such as to prevent superinfection [52]. It is possible that this protein plays a similar role during MOPV or LASV infection, by promoting internalization of the alpha-dystroglycan, which acts as a reservoir for these viruses, to facilitate viral entry steps or limit the cell infection and subsequent over-stimulation of innate immune responses. The cellular RINT1 protein, which promotes human papillomavirus 16 replication, was shown to only interact with LASV Z protein in our Y2H assay [53]. The fact that this protein is known to mediate vesicle budding from the Golgi apparatus to the endoplasmic reticulum (ER) suggest that protein and membrane trafficking are important host-cell processes which could be possibly affected during arenavirus infection.

Importantly, we showed that two HECT-domain E3 ubiquitin ligases of the Nedd4 family, ITCH and WWP1, interact with both LASV and MOPV Z protein. As many RNA viruses are already known to recruit proteins from the Nedd4 family, including ITCH, to facilitate budding, we investigated the role of ITCH, WWP1 and its homolog WWP2 during infection by various old-world arenaviruses [27,54,55,56,57]. We established that ITCH was the candidate whose depletion in cells most severely affected infectious-particle production by the pathogenic LASV, LCMV, and LUJV and the non-pathogenic MOPV. We thus focused on its role. The reduction of infectious-particle production by si-ITCH cells following infection was greater for MOPV than that of the other mammarenaviruses. As the Y2H screening was performed in yeast, it was necessary to retest each physical interaction by immunoprecipitation and colocalization assays in a more relevant mammalian cell line. Immunoprecipitation assays confirmed the interaction between Z protein and ITCH for the previously tested old-world arenaviruses, as well as one new-world arenavirus (JUNV, candid#1 strain), indicating that the interaction with ITCH can be observed in different clades of arenaviruses. Interestingly, less LUJV Z protein was associated with ITCH than that of the other viruses following HA- and FLAG-mediated immunoprecipitation, despite the transfection of similar quantities of Z protein for all conditions. This is in accordance with the moderate loss of infectious-particle production by si-ITCH cells infected with LUJV relative to that of the other mammarenaviruses. This correlation reinforces the relationship between the interaction of Z with ITCH and its ability to promote infectious-particle production.

We further explored the functional role of the Z-ITCH interaction in arenavirus infection by focusing on LASV and MOPV infection, as we have extensively compared these two viruses in previous studies. Viral late domains allow many viruses to promote viral progeny release from the cell by mediating the recruitment of ESCRT proteins [58]. We identified the MOPV and LASV PPxY motif in the C-terminal late domain to be the binding site of ITCH. The PPxY motif is already known to bind Nedd4-like HECT ubiquitin ligases to promote viral budding [59,60,61,62,63]. The PPxY late domain does not seem to bind ESCRT proteins but can directly bind the WW domain, a small globular domain composed of 38–40 semi-conserved amino acids, found in Nedd4 family of E3 ligases [64]. It is thus possible that the PPxY motif of LASV and MOPV Z protein interacts with the WW domain of ITCH and other E3 ubiquitin ligases such as WWP1 and WWP2. A number of reports have indicated that efficient virus budding through the PPxY late domain is dependent on ubiquitination and this dependence can be explained, in part, by the E3 ubiquitin ligase activity of their Nedd4-family binding partners [62,65]. We thus sought to identify if ITCH was responsible for MOPV or LASV Z ubiquitination. Interestingly, in transfection conditions, it appears that the late PPxY domain was necessary for the efficient ubiquitination of MOPV and LASV Z protein by ITCH but ITCH may not be the sole E3 ubiquitin ligase responsible for this modification as Z could still be ubiquitinated in absence of ITCH. Conversely, our results suggested that the MOPV Z protein was not readily ubiquitinated during infection. This observation is in accordance with a recent study in which authors showed that three Nedd4 family proteins (Nedd4, WWP1, and ITCH) bind the PPxY domain of LCMV and JUNV Z protein. Although this interaction resulted in ubiquitination of Z in transfection condition, Z ubiquitination was also dispensable for virus particle release [66].

The fact that ITCH is recruited to the Z PPxY motif raised the possibility that this protein may be recruited by the Z protein to promote the late stages of viral life cycle. We showed that silencing of ITCH in MOPV- or LASV-infected cells impaired the release of infectious virions without affecting the replication steps, confirming that ITCH is involved in the late stages of the MOPV and LASV life cycle. The data appear to be robust, as they were obtained from two different cell lines, HeLa and A549 cells. However, it would be of interest to assess the role of ITCH in a more physiological cell line, such as antigen-presenting cells, especially dendritic cells, as these cells are among the first target cells in vivo [67,68]. In contrast to what we observed here, it has been shown that the PPxY late domain of LCMV is not required for virus budding but instead promotes defective interfering particle release. As MOPV and LASV Z protein contain two late domains, PPxY and PTAP, it appears that the number and structure of such late domains could result in functional differences during viral budding [69]. VLP-budding assays and flow cytometry experiments showed that the interaction with ITCH was important for the release of both MOPV and LASV, as the silencing of ITCH decreased the release of MOPV and LASV VLPs and increased the intracellular level of Z protein in MOPV infected cells. We chose to target ITCH by silencing experiments rather than using E3 ligases activity inhibitor due to the lack of specificity of this method and its possible off-target issues. In our functional analysis of ITCH, the reduction in infectious particle release was greater for MOPV than LASV in si-ITCH transfected cells relative to normal conditions. Moreover, silencing of WWP1 had a greater effect on MOPV than LASV replication. Thus, the MOPV Z protein may be more susceptible to the activity of the Nedd4 family. In addition, Nedd4 has also been shown to bind the LASV Z protein, but such an interaction has not been observed for MOPV [33]. Therefore, Nedd4 and ITCH may have redundant roles during LASV infection. The silencing of ITCH in LASV-infected A549 cells could be, in part, rescued by Nedd4, explaining why the extinction of ITCH expression had a greater effect on MOPV-infected cells. Our conclusions are in line with the work of Ziegler et al., as the silencing of ITCH decreased the proportion of LCMV VLPs released, and a compound preventing PPxY-Nedd4 interaction inhibited both LCMV and LASV Z VLPs release [66,70].

This study increases our understanding of mammarenaviruses infection by completing the interactome of LASV and MOPV Z protein. Our screening reveals a positive role of ITCH for old world mammarenaviruses infectious-particle release. Our study also suggests that if ITCH is able to ubiquitinate LASV and MOPV Z proteins, other E3 ubiquitin ligases may also be involved, and the exact role of this modification in the biology of these viruses remains to be determined.

## Figures and Tables

**Figure 1 viruses-12-00049-f001:**
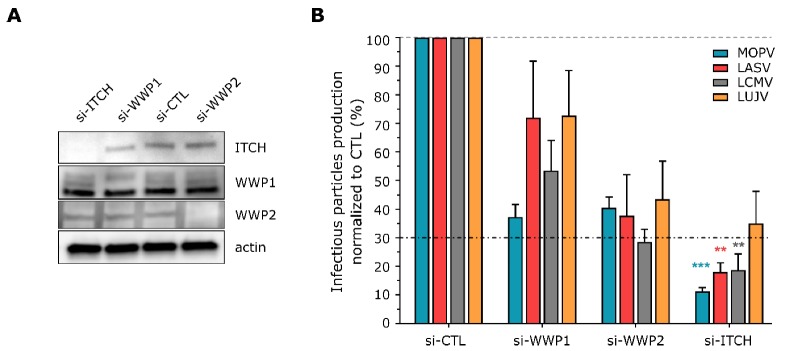
siRNA screen of ITCH, WWP1 and WWP2 during old-world arenavirus infection. (**A**) A549 cells were transfected with siRNA against ITCH (si-ITCH), WWP1 (si-WWP1), WWP2 (si-WWP2) or with a non-targeting siRNA (si-CTL) for 72 h before analysis of silencing efficiency by western blotting (*n* = 4 independent experiments). (**B**) A549 cells were transfected with the indicated siRNA for 72 h before infection with the indicated virus at a MOI of 0.01. Two days after infection, cell supernatants were harvested and titrated on Vero cells. The viral titer was calculated as FFU/mL before normalization to control the conditions. The dotted line corresponds to the threshold at which a mean drop of 70% of infectious particles production was observed. The results show the mean percentages ± SEM pooled from four independent experiments. ** *p* < 0.01, and *** *p* < 0.001 as determined by a Kruskal–Wallis test followed by Dunn’s post hoc test for multiple comparisons.

**Figure 2 viruses-12-00049-f002:**
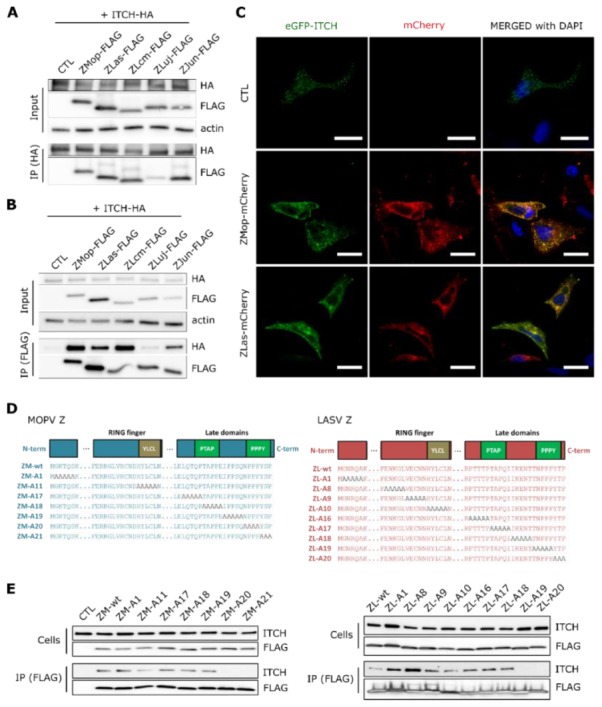
Confirmation of the interaction between various arenavirus Z proteins and the cellular protein ITCH. (**A**) 293T cells were co-transfected with HA-tagged ITCH and/or FLAG-tagged Z protein of MOPV (ZMop-FLAG), LASV (ZLas-FLAG), LCMV (ZLcm-FLAG), LUJV (ZLuj-FLAG), JUNV (ZJun-FLAG). After 15 h, cells were lysed and protein extracts were readily analyzed (input) or incubated with anti-HA magnetic beads for ITCH immunoprecipitation (IP (HA)). FLAG-tagged Z co-immunoprecipitated proteins were detected by western blotting (*n* = 3 independent experiments). (**B**) 293T cells were co-transfected with HA-tagged ITCH and/or FLAG-tagged Z protein of MOPV (ZMop-FLAG), LASV (ZLas-FLAG), LCMV (ZLcm-FLAG), LUJV (ZLuj-FLAG), JUNV (ZJun-FLAG). After 15 h, the cells were lysed (input) or immunoprecipitated with FLAG magnetic beads (IP FLAG) and exogenous HA-tagged ITCH proteins were detected by western blotting (*n* = 3 independent experiments). (**C**) HeLa cells were co-transfected with the indicated plasmids and fixed after 15 h of culture for confocal microscopy. Exogenous eGFP-ITCH is shown in green and the MOPV and LASV Z-mCherry viral proteins are shown in red. Nuclei (blue) were visualized using DAPI reagent and colocalization between eGFP-ITCH and Z-mCherry is shown in yellow. All images were taken on a confocal Zeiss LSM 510 with an Axioscope 63× oil immersion lens objective. Scale bar represents 30 µm. (**D**) Schematic representation of Arenavirus Z matrix protein domains and the generation of mutants by alanine-scanning directed mutagenesis. The scheme represents the position of alanine mutations (in gray) for all MOPV and LASV Z mutants. (**E**) 293T cells were transfected with empty vector, vectors encoding eGFP-ITCH, LASV, or MOPV Z WT FLAG-tagged protein or their relative mutants obtained from directed alanine-scanning mutagenesis experiments. After 15 h of culture, cells were lysed and cell lysates were directly harvested (cells) or incubated with anti-FLAG magnetic beads (IP (FLAG)). ITCH and FLAG-tagged Z proteins were detected by western blotting (*n* = 3 independent experiments).

**Figure 3 viruses-12-00049-f003:**
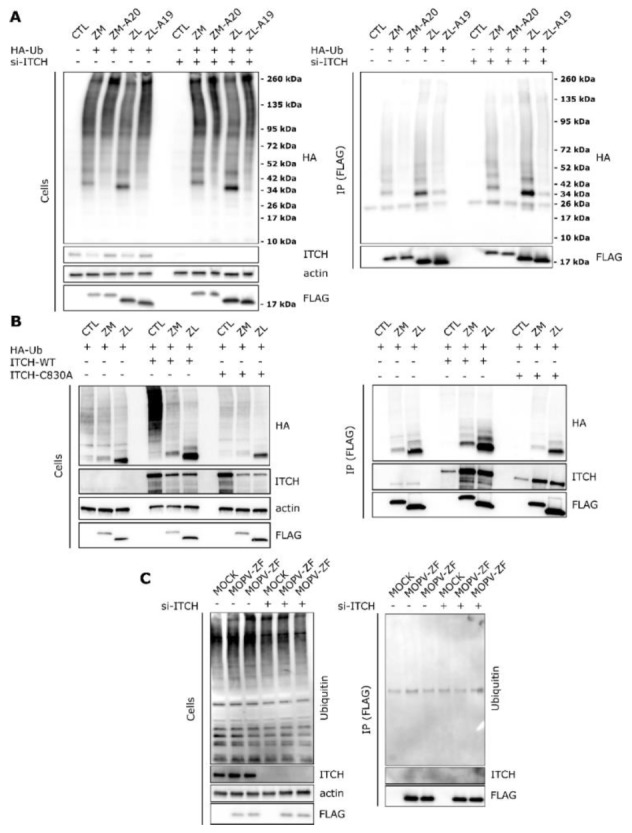
ITCH directly binds to the Z PPxY motif of the LASV and MOPV Z proteins but is not involved in Z ubiquitination. (**A**) The 293T cells were transfected with a non-targeting si-CTL RNA or with si-ITCH for 72 h and then transfected with a HA-tagged ubiquitin and/or with plasmids encoding the FLAG-tagged Z protein of LASV or MOPV (ZL, ZM) or their respective mutants of the late domain (ZL-A19, ZM-A20). A total of 24 h after the second transfection, cells were harvested and total protein extracts (Cells) or proteins co-immunoprecipitated using anti-FLAG magnetic beads (IP (FLAG)) were analyzed by western-blot (*n* = 2 independent experiments). (**B**) The 293T cells were transfected with the wild type ITCH (ITCH-WT) or with the enzymatically inactive mutant ITCH (ITCH-C830A). The same cells were co-transfected with an empty plasmid (CTL), or with the MOPV or LASV Z protein (ZM or ZL). After 15 h of culture, cell extracts were harvested (cells) or incubated with anti-FLAG magnetic beads (IP (FLAG)). Anti-HA, anti-ITCH, anti-FLAG, and anti-actin were used for western blotting analysis (*n* = 2 independent experiments). (**C**) The A549 cells were transfected with the indicated siRNA and cultured for 72 h before being infected with a recombinant MOPV expressing a Z-FLAG tagged protein (MOPV-ZF) at a MOI of 0.1 for 48 h. Cell extracts (cells) were harvested or incubated with anti-FLAG magnetic beads (IP (FLAG)) before western blotting analysis.

**Figure 4 viruses-12-00049-f004:**
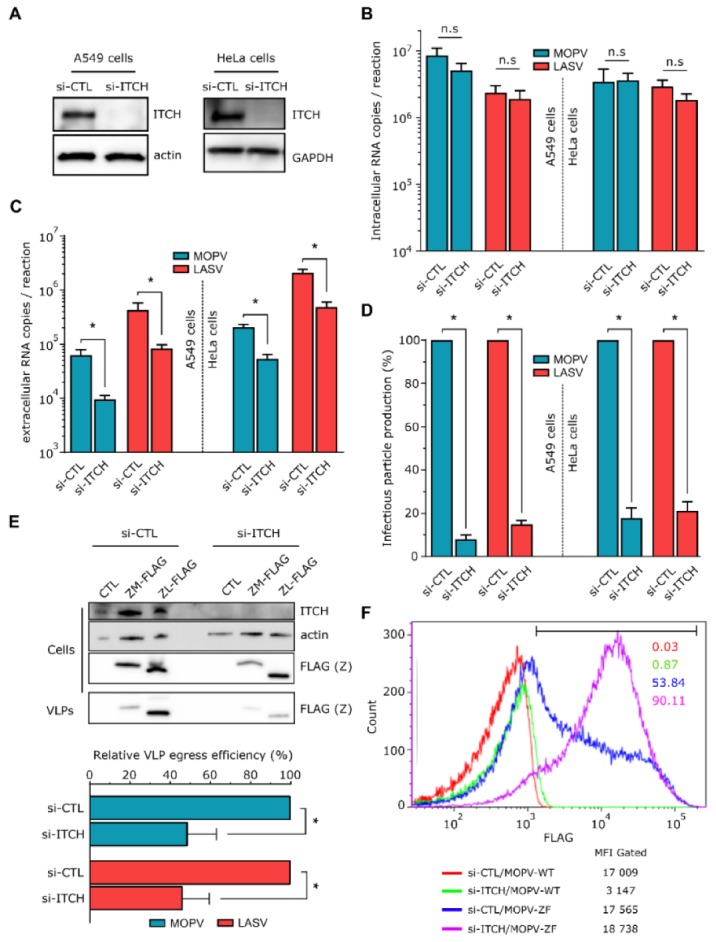
Effect of ITCH on MOPV and LASV infection. (**A**) A549 and HeLa cells were transfected with the indicated siRNA and cultured for 72 h before analysis of silencing efficiency by western blotting (*n* = 4 independent experiments). (**B**–**D**) The same cells as in (**A**) were then infected with LASV or MOPV with a MOI of 0.01 (A549 cells) or with a MOI of 0.1 (HeLa cells) for 1 h before being cultured for two days (A549 cells) or three days (HeLa cells) at 37 °C. Viral RNA was then extracted from the cells (**B**) and supernatants (**C**) for quantification by RTqPCR. Infectious particles from the supernatants were also titrated on Vero cells (**D**). Error bars represent the standard error of the means from four independent experiments. * *p* < 0.05, as determined by the Mann–Whitney test. (**E**) The 293T cells were transfected with the indicated siRNA and cultured for 72 h before being transfected with an empty plasmid (CTL), or with the FLAG-tagged Z protein of MOPV or LASV (ZM-FLAG or ZL-FLAG). After 15 h of culture, cell extracts were directly analyzed by western blotting (cells) and supernatants were harvested, deposited on a sucrose cushion, and ultracentrifuged at 56K rpm for 90 min, before analysis of the pellet by western blotting (VLPs). Representative results are shown, along with a graph representing the intensity of the VLP Z bands. Error bars represent the standard error of the means from four independent experiments. * *p* < 0.05, as determined by the Mann–Whitney test. (**F**) A549 cells were transfected with non-targeting siRNA or ITCH targeting siRNA for 72 h before infection with recombinant MOPV-WT (MOPV-WT) or MOPV-ZFLAG (MOPV-ZF) at a MOI of 2 for 24 h. Intracellular ZFLAG expression was then analyzed by flow cytometry. The bar corresponds to FLAG positive A549 cells. Data shown come from one experiment representative of three replicates.

**Table 1 viruses-12-00049-t001:** Host protein interactors with Lassa virus (LASV) and Mopeia virus (MOPV) Z proteins identified by yeast two-hybrid (Y2H) and validated by gap repair.

Gene Name	Gene ID	Gene Functional Classification	HT-Y2H Screening Hits		Validation by Gap Repair	
			Z MOPV	Z LASV	Z MOPV	Z LASV
*UBQLN4*	56893	Autophagy	9	0	+	
*UBQLN2*	29978	Autophagy	28	0	+	
*UBQLN1*	29979	Autophagy	11	0	+	
*CALCOCO2*	10241	Autophagy	65	0	+	
*OPTN*	10133	Autophagy	7	0	+	
*TAX1BP1*	8887	Autophagy	60	0	+	
*CAPRIN2*	65981	Protein processing	4	13	+	+
*GRIPAP1*	56850	Protein processing	12	0	+	
*AP1G2*	8906	Protein processing	1	5	+	+
*RINT1*	60561	Protein processing	0	8		+
*GOLGA1*	2800	Protein processing	0	2	+	+
*HSP90AA1*	3320	Protein processing	1	0	+	
*WWP1*	11059	Ubiquitination	2	1	+	+
*ITCH*	83737	Ubiquitination	4	2	+	+
*CINP*	51550	Miscellaneous	8	0	+	
*ECD*	11319	Miscellaneous	8	0	+	+
*EPS15L1*	58513	Miscellaneous	20	0	+	
*CCHCR1*	54535	Miscellaneous	3	3	+	+
*KCTD6*	200845	Miscellaneous	0	3		+
*TSG101*	7251	Membrane trafficking	13	36	+	+
*PDCD6IP*	10015	Membrane trafficking	12	30	+	+
*RABEP2*	79874	Membrane trafficking	13	0	+	
*TAB2*	23118	Immunity	3	0	+	
*TLN1*	7094	Cell migration	5	32	+	+

The first and second columns correspond to the canonical gene names and gene IDs, respectively, for the selected interacting cellular proteins. Column 3 indicates the functional role for the selected protein. Columns 4 and 5 show the number of positive yeast colonies obtained for each cellular protein for the indicated virus Z protein (retrieved from mouse brain, human spleen cDNA and human ORFeome libraries). Columns 6 and 7 show the results obtained after the validation step by gap repair in yeast (“+” and blanks indicate the presence and absence of growing colonies, respectively). Tsg101 and PDCD6IP have been already described by others and were used here as a positive control.

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
