# Peer review of "E3 Ligase ITCH Interacts with the Z Matrix Protein of Lassa and Mopeia Viruses and Is Required for the Release of Infectious Particles"

_viruses, 2019, doi:10.3390/v12010049_

Round 1
Reviewer 1 Report
Lassa virus is a rodent-borne virus that can cause hemorrhagic fever in humans after accidental exposure. The Z protein has been established as the driving force of virus particle assembly and budding from the infected cells. To better understand the function of Z protein, the authors sought to identify host proteins that interact with the Z protein of Lassa virus and the closely related Mopeia virus by using Y2H screening. The E3 ligase ITCH, among many other hits, was identified to interact with Z proteins. The interaction was further confirmed in mammalian cells with immunoprecipitation and IFA. The Z proteins seem to be ubiquitinated in transfected cells but not in infected cells. At the same time, ITCH may facilitate virus particle release. Although the interaction between ITCH and LASV Z protein has been recently reported by others, authors provide direct evidence that ITCH has a proviral function during LASV and MOPV infection.
1) Lines 515-519: the main conclusion of the study is that ITCH facilitates the release of LASV particles. Although this is likely based on the VLP data and current knowledge, the conclusion could be strengthened by performing one step growth assay (MOI >1) and assess the intracellular viral protein level (NP or Z ) like that in the VLP assay (Fig 4E) in ITCH-knockdown cells. All infection experiments were performed at low MOI (0.01 or 0.1) as shown in Figure 1 and 4. The supporting data is the similar levels of intracellular virus RNA measured by PCR, which only represents certain species of viral RNA (Fig 4B).
2) Lines 168-170: Authors need to clarify why Co-IP was performed using two antibodies one after the other in the same samples?
3) Line 304: adenine or Alanine?
Author Response
1) Lines 515-519: the main conclusion of the study is that ITCH facilitates the release of LASV particles. Although this is likely based on the VLP data and current knowledge, the conclusion could be strengthened by performing one step growth assay (MOI >1) and assess the intracellular viral protein level (NP or Z) like that in the VLP assay (Fig 4E) in ITCH-knockdown cells. All infection experiments were performed at low MOI (0.01 or 0.1) as shown in Figure 1 and 4. The supporting data is the similar levels of intracellular virus RNA measured by PCR, which only represents certain species of viral RNA (Fig 4B).
We agree and therefore performed a FACS analysis at high MOI to assess the proportion of intracellular Z protein in cells with or without ITCH expression. The main conclusion of this assay strengthen the fact that ITCH facilitates the release of LASV particles. Indeed, the absence of ITCH results in accumulation of Z protein within the cells, confirming that ITCH is involved in the viral life cycle at the budding level. This new result is presented in Fig 4F.
2) Lines 168-170: Authors need to clarify why Co-IP was performed using two antibodies one after the other in the same samples?
The Co-IP were made using one antibody, the anti-FLAG OR anti-HA coupled magnetic beads. The sentence has been modified accordingly.
3) Line 304: adenine or Alanine?
It was a mistake, we replaced adenine by alanine, thank you.
Reviewer 2 Report
General comments:
In “E3 ligase ITCH interacts with the Z matrix protein of Lassa and Mopeia viruses and facilitates the release of infectious particles” Baillet et al. describe a Y2H screen using OWAs Z proteins and found that ITCH was involved in the egress of virus-like particles and the release of infectious progeny viruses. The experiments are well thought out, and the manuscript is a well written and thorough account of these data.
Main comments:
The data were very interesting, clearly involved a lot of work, and effectively identifies the important role that ITCH has in the egress of the OWAs examined here in vitro. The main comment I have regarding the experiments is that it would strengthen the paper greatly if a further set of confirmatory experiments was performed using an ITCH inhibitor, examining whether similar data to the siRNA knock-down was observed, and the therapeutic implications of this (maybe a discussion point if the experiments cannot be carried out).
Also, the western blots are of varying quality throughout the paper, and if possible, should be repeated for the final submission (although again, it’s not necessary to redo any experiments just to obtain protein lysates). Specifically, Fig 1A (WWP2), 2B (FLAG), 2E (FLAG), and 4E (FLAG).
Finally, I think it would be beneficial to the reader if several of the figure legends were expanded on as, at the moment, they are fairly brief, and a lot of the acronyms used are not explained which leaves the reader hunting through the text to find out what sample is what.
Minor comments:
Line 118-119: Please give Genbank numbers for these viruses.
Line 258: Please state in the legend what the dotted line if for in Fig 1B.
Line 309: Fig 2C – very nice images, but is it possible(with the data you have already collected) to have higher-magnification inserts to better show the co-localization?
Author Response
The data were very interesting, clearly involved a lot of work, and effectively identifies the important role that ITCH has in the egress of the OWAs examined here in vitro. The main comment I have regarding the experiments is that it would strengthen the paper greatly if a further set of confirmatory experiments was performed using an ITCH inhibitor, examining whether similar data to the siRNA knockdown was observed, and the therapeutic implications of this (maybe a discussion point if the experiments cannot be carried out).
We agree that using ITCH inhibitors compounds would strengthen the paper. However, to our knowledge, there is no specific ITCH inhibitor commercially available. The drugs that we can get on the market can target the HECT domain of E3 ligases but using them implies having side effects because all E3 ligases enzymes would have been inhibited. We thus preferred to specifically target ITCH by using siRNA. We added a sentence in the discussion section (line 555) :” We chose to target ITCH by silencing experiments rather than using E3 ligases activity inhibitor due to the lack of specificity of this method and its off-target issues.”
Also, the western blots are of varying quality throughout the paper, and if possible, should be repeated for the final submission (although again, it’s not necessary to redo any experiments just to obtain protein lysates). Specifically, Fig 1A (WWP2), 2B (FLAG), 2E (FLAG), and 4E (FLAG).
Unfortunately, we cannot have access to protein lysates of VLP experiments because the whole sample has been loaded for western blot after ultracentrifugation. The samples concerned by immunoprecipitation experiments are also fully loaded for western blots analysis in order to maximize the quantity of co-immunoprecipitated proteins. The blot FIG 1A concerning WWP2 has been redone and has been implemented in the new manuscript.
Finally, I think it would be beneficial to the reader if several of the figure legends were expanded on as, at the moment, they are fairly brief, and a lot of the acronyms used are not explained which leaves the reader hunting through the text to find out what sample is what.
The figures legends are now expanded and all important acronyms are explained.
Minor comments:
Line 118-119: Please give Genbank numbers for these viruses.
The Genbank numbers have been added.
Line 258: Please state in the legend what the dotted line if for in Fig 1B.
The problem is fixed in the legend, thank you.
Line 309: Fig 2C – very nice images, but is it possible (with the data you have already collected) to have higher-magnification inserts to better show the co-localization?
Unfortunately, with the data already collected, we do not have images with “natural” higher magnification. We can numerically zoom into the images, but not without a loss in the pictures resolution.
Reviewer 3 Report
By using yeast two-hybrid screening, Baillet et al. identified several cellular proteins that interact with the Z matrix protein of two Old World mammarenavirus LASV and MOPV. The authors then focused on the interaction of E3 ligate ITCH with Z protein and the function of ITCH on viral egress. Overall, the experiments were well done, and the results support the conclusions. It would be great if the authors could address the following comments to strengthen the manuscript.
Major comments:
(1) Although the authors have demonstrated the interaction of Z and ITCH by co-IP and co-localization experiments in plasmid co-transfected cells, it would be great to demonstrate the interaction of Z and ITCH during MOPV or LASV infection. Although the authors stated in line 358 “while ITCH could interact with MOPV Z protein during infection, as shown by the slight quantity of ITCH co-immunoprecipitated with the Z proteins in si-CTL cells”, the data in Fig 3C is not convincing.
(2) In Figure 4, the authors showed that LASV and MOPV VLP production was affected by the depletion of ITCH expression, indicating that ITCH may involve in the viral release. It would be great to check whether overexpression of ITCH will increase VLP production, which will directly support the authors’ statement “ITCH promotes LASV and MOPV infectious particle production by facilitating viral release”.
Minor comments:
(1) Line 25-26: Please rephrase the sentence to make it clear.
(2) Line 69: LCMV only has one PPxY late domain. Please correct.
(3) Line 123: LASV with a FLAG-tagged Z protein was not used in the manuscript;
(4) Line 268: please change “old world” to “Old-World”;
(5) Page 10, Fig 3A and 3B, it would be great if authors could show the band densities in HA part (both cells and IP) so that readers could easily understand 3.4. section.
Author Response
(1) Although the authors have demonstrated the interaction of Z and ITCH by co-IP and colocalization experiments in plasmid co-transfected cells, it would be great to demonstrate the interaction of Z and ITCH during MOPV or LASV infection. Although the authors stated in line 358 “while ITCH could interact with MOPV Z protein during infection, as shown by the slight quantity of ITCH co-immunoprecipitated with the Z proteins in si-CTL cells”, the data in Fig 3C is not convincing.
We agree that the band is very faint and that the quantity of ITCH co-immunoprecipitated with Z is low. We have tried to increase the signal by performing a new experiment, but the result obtained was similar, with a band of similar intensity. Nevertheless, we think that this result confirms that Z and ITCH interacts during infection conditions, even if only a limited amount of proteins can be detected.
(2) In Figure 4, the authors showed that LASV and MOPV VLP production was affected by the depletion of ITCH expression, indicating that ITCH may involve in the viral release. It would be great to check whether overexpression of ITCH will increase VLP production, which will directly support the authors’ statement “ITCH promotes LASV and MOPV infectious particle production by facilitating viral release”.
In order to respect the deadlines of the journal for resubmission, we won’t be able to perform this experiment, as it would take several weeks to perform these additional experiments. However, we tempered our conclusions with slight modifications. These modifications appear line 3, 86 and 394.
Minor comments:
(1) Line 25-26: Please rephrase the sentence to make it clear.
The sentence has been replaced by the following one: “The PPxY late-domain motif of the Z proteins is required for the interaction with ITCH, although the E3 ubiquitin-ligase activity of ITCH is not involved in Z ubiquitination.”
(2) Line 69: LCMV only has one PPxY late domain. Please correct.
It is a mistake. The sentence has been replaced by this one: “Importantly, the Z proteins of LASV and LCMV contain a PPxY late domain that is known, as for enveloped viruses, to mediate viral budding through the recruitment of the ESCRT-associated protein TSG101 and Alix (25, 26). »
(3) Line 123: LASV with a FLAG-tagged Z protein was not used in the manuscript;
You are right. The problem has been fixed.
(4) Line 268: please change “old world” to “Old-World”; It has been changed, thank you.
(5) Page 10, Fig 3A and 3B, it would be great if authors could show the band densities in HA part (both cells and IP) so that readers could easily understand 3.4. section.
You are right. The protein ladder has been added in figure 3.